# Medication-Related Problems Identified and Addressed by Pharmacists Dispensing COVID-19 Antivirals at a Community Pharmacy

**DOI:** 10.3390/pharmacy11030087

**Published:** 2023-05-20

**Authors:** Danielle Kieck, Leeann Mahalick, Thanh Truc Vo

**Affiliations:** Nesbitt School of Pharmacy, Wilkes University, 84 W South Street, Wilkes-Barre, PA 18766, USA; leeann.mahalick@wilkes.edu (L.M.); thanhtruc.vo@wilkes.edu (T.T.V.)

**Keywords:** community-based pharmacy, medication-related problems, COVID-19

## Abstract

Safe dispensing of coronavirus disease 2019 (COVID-19) oral antivirals requires comprehensive patient assessment to identify and address significant medication-related problems (MRPs). Given the fast-paced environment of community pharmacies and limited access to outside patient records, there are challenges with pharmacists ensuring the safe and appropriate dispensing of these medications. An independent community pharmacy in Pennsylvania developed and implemented a COVID-19 oral antiviral assessment protocol to systematically review all prescriptions dispensed for nirmatrelvir/ritonavir (Paxlovid™) and molnupiravir (Lagevrio™) to identify and address MRPs. A retrospective review was conducted to assess documented MRPs, including significant drug–drug interactions and inappropriate dosing requiring intervention, for prescriptions dispensed from 9 February 2022 to 29 April 2022. Pharmacists identified one or more significant MRPs requiring intervention on 42 of the 54 nirmatrelvir/ritonavir prescriptions (78%) and 0 of the 7 molnupiravir prescriptions. Most pharmacist interventions involved drug–drug interactions between nirmatrelvir/ritonavir and HMG-CoA reductase inhibitors and calcium channel blockers, along with four renal dose adjustments for nirmatrelvir/ritonavir. This study highlights the ability of community pharmacists to identify and address MRPs and promotes the use of a protocol to encourage safe dispensing practices for medications prone to MRPs.

## 1. Introduction

The coronavirus disease 2019 (COVID-19) pandemic has highlighted the need for safe and effective oral antiviral therapy for high-risk patients in the community setting [1,2]. Since the pandemic was declared in March 2020, over 100 million positive cases have been documented in the United States [3]. At the beginning of the pandemic, only hospitalized patients meeting certain criteria had the opportunity to receive COVID-19 therapeutics under an emergency use authorization (EUA) [4]. However, even mild-to-moderate COVID-19 illnesses have been associated with cardiovascular or pulmonary damage, as well as the development of “long COVID” symptoms [5]. For many months, the only therapeutic option for nonhospitalized patients with mild-to-moderate symptoms at high risk of progression to severe disease was intravenous monoclonal antibody therapy [4]. Fortunately, in December 2021 the Food and Drug Administration (FDA) authorized the use of oral nirmatrelvir/ritonavir and molnupiravir for patients with mild-to-moderate symptoms at high risk of progression to severe COVID-19 disease [6,7]. Dispensing of these antivirals presented a new opportunity for community pharmacists to utilize their extensive medication knowledge and skills.

Although these oral medications are a groundbreaking addition to the arsenal of COVID-19 therapeutics, they should be dispensed cautiously, because they require a thorough evaluation of patient information to ensure safety [8]. This is especially important for nirmatrelvir/ritonavir, which has numerous and significant drug–drug interactions (DDI) [6]. As a result, collecting a detailed health and medication history for each patient is crucial. COVID-19 oral antiviral prescriptions may be filled at a pharmacy where the patient is not established due to specific government allocation [9]. Moreover, as discovered by Polinski et al., elderly patients on multiple medications are more likely to fill their prescriptions at more than one pharmacy [10]. As distribution continues to become widespread, more community pharmacists are faced with the challenge of obtaining this information to ensure that patients receive safe and appropriate medication therapy [8].

A *Morbidity and Mortality Weekly Report* from June 2022 found that the dispersion of COVID-19 oral antiviral agents began to substantially increase in March 2022, after a series of strategies were implemented to expand COVID-19 oral antiviral access, including the launch of the test-to-treat initiative [2]. The essential role of the community pharmacist in assisting with the management of COVID-19 oral antiviral agents was highlighted during this time, as an estimated 87% of the dispensing sites were pharmacies [2]. The pharmacist role was further expanded in July 2022, when the FDA authorized state-licensed pharmacists to prescribe nirmatrelvir/ritonavir under a revised EUA [11]. Community pharmacists in the United States are the most accessible healthcare professionals, with 96.5% of people living within 10 miles of a community pharmacy, and over 2800 community pharmacies in federally recognized underserved communities [12,13]. This accessibility to patients became an essential factor for the emerging role of pharmacists beyond the traditional dispensing of medications. Since these oral antivirals need to be initiated within a short time frame, the authorization to prescribe nirmatrelvir/ritonavir demonstrates community pharmacists’ vital role in increasing access to care and managing acute illnesses.

The surge in COVID-19 and the increased volume of prescriptions for oral antiviral agents has resulted in pharmacists spending more time coordinating communication between providers and patients. Time constraints, staffing shortages, and lack of access to a patient’s comprehensive medical history represent obstacles that may lead to unintentional medication errors [14]. Recognizing this, the study pharmacy implemented a systematic approach for its pharmacists to assess each COVID-19 oral antiviral prescription. The primary objective of this study was to describe the impact of a pharmacist-driven COVID-19 oral antiviral assessment protocol by quantifying and describing medication-related problems identified and resolved for COVID-19 oral antiviral prescriptions.

## 2. Materials and Methods

This study took place at a family-owned, multigenerational, independent pharmacy located in Pennsylvania. The pharmacy is open 7 days a week and fills approximately 3500 prescriptions per week. Services provided include, but are not limited to, medication synchronization, delivery, medication therapy management (MTM), and immunizations.

A retrospective review of patient information was conducted to assess documented MRPs through the use of the protocol developed by the pharmacy, including significant DDIs and inappropriate dosing requiring intervention, for prescriptions dispensed from 9 February 2022 to 29 April 2022. The study was reviewed and determined as exempt by Wilkes University Institutional Review Board. All patients who received a COVID-19 oral antiviral during the study period were included in the review. Patients were excluded if the prescription for an oral antiviral was received but never dispensed by the pharmacy.

The COVID-19 oral antiviral protocol was developed to address prescribing errors and pharmacy assessment inconsistencies, allowing for the development of a stronger, team-based collaboration between pharmacy staff and local prescribers. Resources that were utilized in the establishment of the protocol to identify and manage DDIs included readily available drug information tools such as Lexicomp or Micromedex, the National Institutes of Health (NIH) website, the EUA fact sheets, and the Liverpool COVID-19 Drug Interactions website [4,6,7,15].

The protocol entailed the use of a checklist (Table 1 and Table 2) that was developed by pharmacists to perform a comprehensive drug utilization review (DUR) for all oral antiviral prescriptions received prior to dispensing. Because multiple patients who filled oral antivirals at the pharmacy were not established patients, the developed checklist served as an additional safety net for pharmacists to ensure that they were appropriately dispensing each antiviral prescription and providing proper patient counseling. Additionally, pharmacists and student pharmacists developed educational handouts for pharmacists, support staff, and local providers (Appendix A and Appendix B). Pharmacist interventions discovered through the use of the checklist were defined as any medication therapy intervention made regarding inappropriate dosing or a significant DDI. A significant DDI was defined as an interaction that required concomitant medications to be held, dose-reduced, or closely monitored. The computer system at the pharmacy was unable to identify DDIs for patients who were not established at the pharmacy. Additionally, the nirmatrelvir/ritonavir EUA fact sheet is not entirely inclusive of medications that may interact with nirmatrelvir/ritonavir and other resources varied on their recommendations. For these reasons, pharmacists utilized at least two resources to check for drug interactions.

Pharmacists used the checklist in Table 1 to verify the date of symptom onset, high-risk condition(s), renal function (eGFR), and current medication list for nirmatrelvir/ritonavir prescriptions. Since many patients who filled nirmatrelvir/ritonavir prescriptions were not established patients, the pharmacist would often contact the provider to receive verbal confirmation or a faxed copy of this information. To ensure no delays in treatment, patients could also provide the necessary information if a provider could not be reached promptly. Patients were also asked about their use of nonprescription medications and/or herbal supplements. If an error or significant DDI was identified, the pharmacist would contact the prescriber to discuss alternative COVID-19 treatment or provide a recommendation to the prescriber and/or patient on how to manage the DDI. For molnupiravir prescriptions, pharmacists utilized the checklist in Table 2 to verify the patient’s date of symptom onset, high-risk condition(s), and pregnancy status. During medication counseling with the patient, pharmacists counseled on the importance of proper contraception during and after treatment with molnupiravir due to reproductive toxicity concerns. The pharmacy provided patient-friendly educational handouts for each patient receiving an oral antiviral in addition to the required EUA fact sheet.

## 3. Results

Sixty-seven prescriptions were sent to the pharmacy for COVID-19 oral antivirals during the study timeframe. Fifty-four patients received nirmatrelvir/ritonavir and seven patients received molnupiravir. Six patients were excluded from the study, as a COVID-19 oral antiviral prescription was sent to the pharmacy but never dispensed. Some reasons that prescriptions were not dispensed include failure of the patient to pick up the prescription or failure to meet the emergency use criteria (i.e., asymptomatic patients). Of the 61 patients evaluated, only 7 were established patients of the pharmacy. Included patients had an average age of 61 years and 62% were female (Table 3). All patients had at least one qualifying factor defined by the Centers for Disease Control (CDC) that put them at high risk of progressing to severe COVID-19 disease [16].

Of the 54 nirmatrelvir/ritonavir prescriptions dispensed, 4 prescriptions (7%) were sent with incorrect dosing in the setting of renal impairment (eGFR < 60 mL/min/1.73 m^2^). In total, 42 nirmatrelvir/ritonavir prescriptions (78%) contained at least one significant DDI. Table 4 highlights key MRPs identified by pharmacists for nirmatrelvir/ritonavir and the management steps taken. Only prescriptions with significant MRPs that required pharmacist intervention were included in the totals. The most common DDI encountered (28 prescriptions) was between nirmatrelvir/ritonavir and HMG-CoA reductase inhibitors (statins). Both simvastatin and lovastatin are contraindicated with concurrent use of nirmatrelvir/ritonavir and must be stopped 12 h before the first dose of nirmatrelvir/ritonavir due to the increased risk of myopathy and rhabdomyolysis [6]. Other statins include instructions to hold the medication throughout therapy and for up to 3 days after finishing nirmatrelvir/ritonavir [15]. Nirmatrelvir/ritonavir drug interactions are complex, and the provider must weigh the risk vs. benefit of holding a concomitant medication, continuing the concomitant medication with increased monitoring, or utilizing alternative antiviral therapy (i.e., molnupiravir). Other common DDIs that required a pharmacist’s intervention were calcium channel blockers (10 prescriptions), fluticasone nasal spray (10 prescriptions), and glucocorticoids (6 prescriptions). 

There were no pharmacist interventions made on any of the seven molnupiravir prescriptions. Due to its second-line role in therapy, molnupiravir only comprised 11% of the total COVID-19 oral antiviral prescriptions dispensed during the study timeframe [4]. Since molnupiravir does not require renal dose adjustments and lacks significant cytochrome P (CYP) 450 or p-glycoprotein (P-gp) metabolism, pharmacist interventions were expected to be minimal [7]. Of the patients who received molnupiravir, all three female patients were not of child-bearing age, eliminating any reproductive concerns, and males were counseled on reproductive risk and using barrier methods during intercourse.

## 4. Discussion

Global healthcare systems experienced a substantial strain during the COVID-19 pandemic due to healthcare worker shortages and increased burnout rate as the pandemic progressed [17]. Staffing shortages and limited appointment availability contributed to decreased utilization of healthcare services during the pandemic [17]. Thus, the COVID-19 pandemic had a significant impact on the provision of pharmaceutical care in the community setting [18]. Pharmacies had to adapt to changes in workflow and expand on their pre-existing roles to provide novel services [19,20,21]. As demonstrated by this study, a new role for pharmacists during the COVID-19 pandemic was ensuring safe dispensing of COVID-19 antivirals.

Implementation of a pharmacist-driven COVID-19 antiviral assessment protocol was associated with numerous pharmacist interventions regarding dosing and/or significant DDIs with nirmatrelvir/ritonavir. No pharmacist interventions were made on the molnupiravir prescriptions received during the specified time frame. Since these medications are under EUA and have only been available for a short period of time, they are prone to both prescribing and dispensing errors [22,23]. The National Institute for Safe Medication Practices (ISMP) issued warnings about nirmatrelvir/ritonavir prescribing errors, including standard dosing for patients with renal impairment, prescribing the incorrect quantity, providing directions to take two tablets twice a day in conflict with instructions included on the packaging, and not checking renal function or concomitant medications for drug interactions [23]. Factors contributing to these errors included provider unawareness of renal dosing and a need for drug interaction screening, as well as confusion about available packaging and the number of tablets per regimen [23]. Based on the results of this study, using a protocol and a systematic approach to reviewing all COVID-19 antiviral prescriptions in close collaboration with local providers supports safe medication dispensing practices. 

For patients filling a prescription for nirmatrelvir/ritonavir, the pharmacist should ideally request confirmation of a positive COVID-19 test, date of symptom onset, high-risk qualification(s), most recent eGFR, and a medication list, including nonprescription medications and herbal supplements. If filling a nirmatrelvir/ritonavir prescription for a child ≥12 years old, the pharmacist should ensure that the patient weighs at least 40 kg [4]. The study results highlight the importance of implementing a systematic approach to dispensing COVID-19 oral antivirals in community pharmacies, especially for nirmatrelvir/ritonavir. Additionally, it emphasizes the ability of pharmacists to collect pertinent information on which to base safe medication therapy, and supports pharmacists prescribing COVID-19 therapies through the growing test-to-treat initiative.

Although various tools and checklists have been created since the COVID-19 oral antivirals were authorized for use, this study was completed at a time when resources were limited. Pharmacist recommendations/interventions were made based on the available evidence at the time the study was completed. In early June 2022, the FDA created a checklist for providers to utilize in conjunction with other resources to ensure the safe prescribing of nirmatrelvir/ritonavir and molnupiravir and to serve as an aid to clinical decision making [24,25]. The FDA checklists were developed primarily for prescriber use; however, pharmacists may find these checklists beneficial when assessing antiviral prescriptions for appropriateness during the dispensing process, as they contain similar information to the checklists used within this study [24]. To date, there are no available studies describing the impact of the FDA checklists on prescribing errors or other MRPs.

According to the Food and Drug Administration (FDA), more than 100,000 medication errors are reported each year, and impact approximately 7 million U.S. patients per year [26]. In the community setting, about 1.5% of all prescriptions dispensed will have a dispensing error [26]. One type of dispensing error is the failure of a pharmacist to catch a drug interaction, often due to a lack of access to or incomplete patient health records. Thus, a collaboration between prescribers, nurses, and pharmacists is essential in identifying medication discrepancies that can result in patient harm [27]. In a time when staff shortages were affecting healthcare nationwide, communication and collaboration were crucial during the initial dispensing period of the oral antivirals [28]. While the long-established relationships between the pharmacy and some of the local prescribers made it easier for the pharmacists to make recommendations and request information, effective communication and fluidity were necessary by the healthcare team to ensure the patient received prompt treatment. This interdisciplinary collaborative approach enabled providers to prescribe oral antivirals with ease of mind, knowing that the pharmacy would provide a second safety check before the medication was dispensed. From the pharmacy’s perspective, pharmacists can promptly receive the essential health information they need from the prescriber’s office to dispense the medications. 

In addition to decreasing dispensing errors, the interdisciplinary collaboration between pharmacists and providers allows pharmacists to intervene on other medication errors, such as prescribing errors. A systematic review and meta-analysis on pharmacist-led educational interventions for reducing medication errors found that educational programs led by a pharmacist were associated with significant reductions in the overall rate of medication error occurrence [29]. Because nirmatrelvir/ritonavir and molnupiravir were just recently authorized for emergency use, prescribers had to quickly learn about the nuances associated with these new medications and adapt their office workflow to meet the requirement of the pharmacy’s COVID-19 antiviral dispensing protocol. To assist in the education process, the pharmacy had a pharmacist visit some local prescribers during the initial period when the oral antiviral agents were first made available and provided educational handouts regarding the medications.

The pharmacy constantly adapted its workflow during the time period that it dispensed the antivirals, refining the process to be more time-efficient for both the pharmacy and the prescriber office. An enhancement made as time progressed was the establishment of a formalized document for pharmacists to fax prescribers, enabling prescribers to quickly and efficiently fill out the information needed for dispensing the antivirals (Appendix C). Every time the pharmacy received a prescription for an oral antiviral without necessary information (i.e., date of symptom onset, positive COVID-19 test, renal function, and medication list) a pharmacist or technician would fax a form back to the prescriber’s office to be completed. This implementation reduced the time pharmacists spent calling the prescribing office and lowered the risk of incomplete information sent by prescribers. Other ways in which the pharmacy adapted its workflow included offering curbside pick-up for individuals receiving an oral antiviral agent to minimize exposure risk. Furthermore, patients were notified to not come to the pharmacy until they received a call from the pharmacy staff that their prescriptions were ready for pick up. 

Although this study successfully describes numerous MRPs and clinical interventions made on nirmatrelvir/ritonavir prescriptions, there were some limitations to this analysis. The results show interventions pharmacists identified and acted upon through contacting the patient and/or provider; however, if a DDI was present, pharmacists were not consistent in documenting if a patient was already told by their provider to hold or reduce the dose of a concomitant medication before they made their recommendations. Therefore, the results do not show the failure of the provider to address DDIs, but rather the ability of the pharmacist to identify potential DDIs. Another limitation was that the sample size analyzed was small since the pharmacy only received a limited number of antiviral prescriptions during the study time frame. National prescribing of the antivirals was not as prevalent shortly after EUA was granted, likely accounting for the limited sample size [2]. Antiviral dispensing began to rise substantially after the conclusion of our study due to initiatives that increased access to the medications [2]. Additionally, underutilization of COVID-19 antivirals remains a concern, and may have also contributed to the limited sample size. A national poll of patients infected with COVID-19 from May to July 2022 revealed that only 11% of patients were prescribed antiviral therapy [30]. The time frame of this study was relatively short, at approximately three months; however, this captures the time frame shortly after both antivirals received EUA and before educational resources became widespread. These factors may have contributed to the numerous MRPs identified. Lastly, follow-up with patients who required intervention was not completed, meaning safety during or after therapy could not be assessed. Despite this, interventions made prior to dispensing of the medications likely prevented unintended side effects from inappropriate dosing or DDIs, supporting the potential for safer dispensing through the use of a systematic assessment protocol.

## 5. Conclusions

The protocol developed by the pharmacy has provided its staff with a systematic approach to evaluating each prescription for nirmatrelvir/ritonavir and molnupiravir, leading to numerous clinical interventions. The results of this study emphasize the importance of a team-based approach to care for patients being treated for COVID-19 in the outpatient setting. Providers and pharmacists should be in communication regarding concomitant medication adjustments or other safety precautions being recommended. As supported by the results of this study, dispensing COVID-19 antivirals offers an opportunity for community pharmacists to utilize their knowledge and medication information skills to ensure that patients are receiving safe medication therapy.

## Figures and Tables

**Table 1 pharmacy-11-00087-t001:** Nirmatrelvir/ritonavir (Paxlovid) pharmacy assessment checklist.

Nirmatrelvir/Ritonavir (Paxlovid): Pharmacy Assessment		
1	Date of symptom onset (Must be within 5 days of filling)	
2	High-risk condition(s) (Patients must have one or more high-risk conditions defined by the CDC)	
3	Renal Function (eGFR) and Hepatic Function	
4	Medication List (including over-the-counter and herbals)	
5	Pharmacy Interventions (List any pharmacist interventions made)	

**Table 2 pharmacy-11-00087-t002:** Molnupiravir (Lagevrio) pharmacy assessment checklist.

Molnupiravir (Lagevrio): Pharmacy Assessment	
1	Date of symptom onset (Must be within 5 days of filling)
2	High-risk condition(s) (Patients must have one or more high-risk conditions defined by the CDC)
3	Pregnancy Status (Must not be actively pregnant)
4	Pharmacy Interventions (List any pharmacist interventions made)

**Table 3 pharmacy-11-00087-t003:** Demographics.

Demographics	n = 61
Sex	
Male	23 (38%)
Female	38 (62%)
Age	
18–29	2 (3%)
30–49	6 (10%)
50–64	23 (38%)
65 and older	30 (49%)
Number of Concomitant Medications *	
0–4	14 (23%)
5–9	15 (25%)
10 or greater	21 (34%)

* Researchers were unable to obtain full medication lists in the retrospective chart review for 2 nirmatrelvir/ritonavir and 5 molnupiravir prescriptions due to inconsistencies in documentation, but were able to obtain documented interventions for specific medication therapy problems. Furthermore, it is important to note that workflow for molnupiravir prescriptions did not require pharmacists to verify a full medication list due to no significant documented drug–drug interactions.

**Table 4 pharmacy-11-00087-t004:** Summary of key pharmacist interventions on nirmatrelvir/ritonavir (Paxlovid) prescriptions *.

Medication-Related Problem	Number of Prescriptions **	Pharmacist Intervention(s)
DDI—Statins	28	Pharmacists recommended statins be held for the duration of nirmatrelvir/ritonavir therapy and 3 days post-treatment ***.
DDI—Calcium Channel Blockers (CCB)	10	Pharmacists recommended a 50% dose reduction of certain CCBs, such as amlodipine, and counseled patients to closely monitor blood pressure and heart rate for the duration of nirmatrelvir/ritonavir therapy.
DDI—Fluticasone Nasal Spray	10	Pharmacists recommended the patient hold fluticasone for the duration of nirmatrelvir/ritonavir therapy.
DDI—Benzodiazepines	6	Pharmacists recommended reducing as-needed alprazolam dosing by 50% and to monitor for increased drowsiness and sedation for duration of nirmatrelvir/ritonavir therapy.
DDI—Glucocorticoids	6	Pharmacists contacted providers to discuss the benefits and risks of using a glucocorticoid (dexamethasone, prednisone) outpatient for COVID-19, as current guidelines recommend against this.
DDI—Antiplatelets	4	Pharmacists called the provider/cardiologist to discuss the benefits and risks associated with decreased or increased antiplatelet efficacy when used together with nirmatrelvir/ritonavir.
Dose too high—Dose Adjustment for eGFR ≥ 30 to <60	4	Pharmacist contacted the provider and had prescription directions changed to 150 mg nirmatrelvir (one 150 mg tablet) with 100 mg ritonavir (one 100 mg tablet) twice daily for 5 days.
Other	4	PDE-5 inhibitors—pharmacists counseled patients to hold as-needed PDE5 for the duration of nirmatrelvir/ritonavir therapy. Opioids—Pharmacists counseled patients to limit the use of as-needed opioids for the duration of nirmatrelvir/ritonavir therapy and to monitor for increased drowsiness and sedation.
DDI—Alpha-Blockers	3	Pharmacists recommended holding tamsulosin and alfuzosin for the duration of nirmatrelvir/ritonavir therapy.

* Recommendations were made based on available evidence and resources during the timeframe analyzed in the study. Recommendations may have changed since the completion of this study. ** While 42 of the 54 prescriptions had interventions, some of the prescriptions had multiple interventions. DDI = Drug–drug interaction. *** After this study concluded nirmatrelvir/ritonavir product labeling updated to recommend holding simvastatin 12 hours before and 5 days after treatment.

## Data Availability

The data presented in this study are available on request from the corresponding author.

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
