# Peer review of "Medication-Related Problems Identified and Addressed by Pharmacists Dispensing COVID-19 Antivirals at a Community Pharmacy"

_pharmacy, 2023, doi:10.3390/pharmacy11030087_

Round 1

Reviewer 1 Report

Thank you for the opportunity to review your manuscript! The topic is novel, I could not find many previously published studies in databases, and is within the scope of the journal. Furthermore, the manuscript has fine English language, it is easy to read.

My main concern regarding this manuscript, is the very limited data material (N=61) sampled from only one pharmacy. Furthermore, the C-19 vaccine checklists used were developed and used by the study pharmacy only. Therefore, the significance of outcome is low, by my opinion. How do you consider validity in you results? In what way does these checklists differ from the FDA checklists? Would it be feasible to include data on MRPs identified by pharmacists using the FDA checklists as well?

Other comments:

The study lacks information on research ethics, and is needed since personal and health data has been collected. Does the project have some kind of ethical revision and approval?

Result presentation:

* it would be nice to have a flow diagram of data inclusion

* include a table of patients characteristics, e.g., information on age, gender, use of medicines, and other relevant information, if available

* Table 3: Could consider changing the order of the listed MRPs, listing them according to number of identified prescriptions (from highest to lowest)

Author Response

Thank you for reviewing our paper, and we really appreciate your comments. Below are our responses to your comments:

  1. My main concern regarding this manuscript, is the very limited data material (N=61) sampled from only one pharmacy. Furthermore, the C-19 vaccine checklists used were developed and used by the study pharmacy only. Therefore, the significance of outcome is low, by my opinion. How do you consider validity in you results? In what way does these checklists differ from the FDA checklists? Would it be feasible to include data on MRPs identified by pharmacists using the FDA checklists as well?

We recognize there may be concern for the limited sample size from one pharmacy and understand this is a limitation to the study. With this being said we submitted to community pharmacy research and quality improvement special edition as we feel this information is still important to share with the pharmacy community as there is lacking research when it comes to community pharmacist roles with assessing oral COVID-19 antiviral therapy.

It is important to note that this study occurred during the time frame shortly after the COVID-19 oral antivirals received emergency use authorization and prescribing of the antivirals was not as prevalent, accounting for the limited sample size.  Additionally, we recognize that data was obtained from a single pharmacy. Use of protocols or other available resources may not be as easily implemented at other pharmacies with different workflow. Despite this, patient safety should be prioritized, and it is the responsibility of a pharmacist to identify medication-related problems on all prescriptions dispensed. Therefore, other pharmacies are encouraged to implement available tools to assess COVID-19 antiviral prescriptions for medication related problems.  We also recognize that the checklist was developed and used by our pharmacy only. However, the information included on our checklist focuses only on relevant, drug-specific criteria that is unlikely to change. Checklists that were developed after completion of our study contain similar criteria, therefore adding external validity in our results. We are unable to identify other studies that utilize resources such as the FDA checklist to identify medication related problems.

In the manuscript we added a discussion regarding the limitation of sample size in the last discussion paragraph (page 8 line 337-344). Also, discussion about the FDA checklists and current studies available was added to the top of page 7 (line 270-274)

  1. The study lacks information on research ethics and is needed since personal and health data has been collected. Does the project have some kind of ethical revision and approval?

This study was reviewed by the Wilkes University Institutional Review board and considered exempt under according to 45 CFR 46.104 (d)(4).  This was a retrospective chart review, and no personal identifiable information was collected or stored, also all the information was reported in aggregate. All researchers on this study completed IRB training and maintained strict ethical standards.

  1. Result presentation: it would be nice to have a flow diagram of data inclusion; include a table of patient characteristics, e.g., information on age, gender, use of medicines, and other relevant information, if available; Table 3: Could consider changing the order of the listed MRPs, listing them according to number of identified prescriptions (from highest to lowest).

Based on this study being a smaller retrospective chart review with one inclusion/exclusion criteria we feel a flow chart is unnecessary and would add additional length to the paper that would not enhance the readability. We did add a demographics table with age, gender and number of concomitant medications (table 3). We did change the order of the content in table 3 (now table 4) according to number of identified prescriptions and feel this make the data more readable.

Reviewer 2 Report

Thank you for giving me the opportunity to read and comment a report “Medication-Realted Problems Identified and Addressed by Pharmacist Dispensing COVID-19 Antivirals at a Community Pharmacy”, by Kieck D, et al.

In the reviewed manuscript, the medication-related problems (MRPs) identified and resolved for COVID-19 oral antiviral prescriptions, has been evaluated.

This paper is well written, correctly structured with a suitable research concept, and it is of relevance to readers of the journal. However, I have a few comments to make below.

·         Author affiliation is incomplete and not properly formatted.

·         The description of the study is usually included at the beginning of the "Materials and Methods" section.

·     The footnote to Table 3 should include the meaning of the acronym DDI.

·         The study period is less than 3 months, which may be a limitation that authors should reflect in the manuscript.

·    Finally, it would be advisable to review the bibliography, since the references do not follow the format established by the journal.

Author Response

Thanking for reviewing our article and providing feedback. Here is how we addressed your comments in our paper.

  1. Author affiliation is incomplete and not properly formatted.

Edits to the author affiliation have been made to ensure completeness and proper formatting. Also, Since the research was conducted while all authors were at Wilkes University the research team decided it update the affiliations for all members to reflect this.

  1. The description of the study is usually included at the beginning of the "Materials and Methods" section.

The paragraph containing the description of the study was moved to the beginning of the “Materials and Methods” section.

  1. The footnote to Table 3 should include the meaning of the acronym DDI.

A comment was added to the footnote of table 3 defining the acronym DDI. Note this is now table 4 as we added a demographics table.

  1. The study period is less than 3 months, which may be a limitation that authors should reflect in the manuscript.

Although this short time frame may be viewed as a limitation, it is important for readers to recognize that the time frame studied was shortly after both antiviral medications received emergency use authorization (EUA). This is important because prescribing and dispensing of these novel medications was particularly challenging before additional tools and resources became widespread (such as the FDA checklist). This study highlights the ability of a community pharmacist to effectively acquire and assess information as well as identify medication-related problems with COVID-19 antiviral prescriptions. The results of our study show that medication related problems regarding COVID-19 antiviral prescriptions were numerous shortly after EUA was granted when resources were limited. With various tools and resources now available, pharmacies across the United States are encouraged to utilize these given the amount of MRPs identified utilizing a similar tool at our pharmacy.

This was further described in the last paragraph in the discussion (page 8 line 342-344).

  1. Finally, it would be advisable to review the bibliography, since the references do not follow the format established by the journal.

The bibliography was reviewed and updated to ensure consistency with the format established by the journal.

Round 2

Reviewer 1 Report

Thank you for the revised manuscript and response to my comments.

1)    I find that you have added information on the FDA checklists, and have expanded the discussion on the limited sample size. Thank you, that will help the reader to understand the strengths and limitation of this study. The sample size is still small, however, I find your arguments reasonable, and this study can be of interest for the pharmacy community. This study is showcasing the opportunities a community pharmacy has to use its expertise to make an impact for medication safety. It is admirable that a community pharmacy took the initiative, developed and implemented a strategy for patient safety during the busy months of C-19 pandemic.

2)    Regarding research ethics, I was not familiar with the term “45 CFR 46.104 (d)”, is that specific to the US? However, I "Googled" it, and found that it means code of federal regulations relating to protection of human subjects in research.

3)    I accept the argument regarding flow chart. Thank you for adding a demographic table and revising Table 4. I was wondering about the categorization of age as uneven categories in Table 3, is there a good reason? I guess presenting age as a continuous variable would be a safe choice, reporting mean or median, standard deviation or range. You present average age in the text.

Author Response

Thank you for your comments. 

1) I appreciate the comments you made in edit round one to help strengthen our discussion and am happy to hear you feel it better helps the reader understand the limitations of the study

2) It was human subjects but exempt due to the retrospective, de-identified nature of the review. This was reviewed by the wilkes IRB and a consent waiver was obtained. 

3) The demographic's table was created by looking at some online examples. I personally like how readers can easily see how many patients are 65 and older, as this population is more likely to be high risk and prone to MRPs.